# Inflammasome, the Constitutive Heterochromatin Machinery, and Replication of an Oncogenic Herpesvirus

**DOI:** 10.3390/v13050846

**Published:** 2021-05-06

**Authors:** Sumita Bhaduri-McIntosh, Michael T. McIntosh

**Affiliations:** 1Division of Infectious Disease, Department of Pediatrics, University of Florida, Gainesville, FL 32610, USA; 2Department of Molecular Genetics and Microbiology, University of Florida, Gainesville, FL 32610, USA; 3Child Health Research Institute, Department of Pediatrics, University of Florida, Gainesville, FL 32610, USA

**Keywords:** inflammasome, NLRP3, caspase 1, heterochromatin, constitutive heterochromatin, epigenome, epigenetics, KRAB–ZFP, KAP1, TRIM28, TIF1β, gammaherpesvirus, herpesvirus, Epstein–Barr virus, lytic phase, reactivation, ATM, viral protein kinase, ZEBRA, lytic switch

## Abstract

The success of long-term host–virus partnerships is predicated on the ability of the host to limit the destructive potential of the virus and the virus’s skill in manipulating its host to persist undetected yet replicate efficiently when needed. By mastering such skills, herpesviruses persist silently in their hosts, though perturbations in this host–virus equilibrium can result in disease. The heterochromatin machinery that tightly regulates endogenous retroviral elements and pericentromeric repeats also silences invading genomes of alpha-, beta-, and gammaherpesviruses. That said, how these viruses disrupt this constitutive heterochromatin machinery to replicate and spread, particularly in response to disparate lytic triggers, is unclear. Here, we review how the cancer-causing gammaherpesvirus Epstein–Barr virus (EBV) uses the inflammasome as a security system to alert itself of threats to its cellular home as well as to flip the virus-encoded lytic switch, allowing it to replicate and escape in response to a variety of lytic triggers. EBV provides the first example of an infectious agent able to actively exploit the inflammasome to spark its replication. Revealing an unexpected link between the inflammasome and the epigenome, this further brings insights into how the heterochromatin machinery uses differential strategies to maintain the integrity of the cellular genome whilst guarding against invading pathogens. These recent insights into EBV biology and host–viral epigenetic regulation ultimately point to the NLRP3 inflammasome as an attractive target to thwart herpesvirus reactivation.

## 1. A Two-Population Problem

Herpesviruses persist in a quiescent/latent state but can switch into the productive/lytic phase to infect new hosts. Though ensuring persistence in the population, this productive phase is a potential dead end in the host, as the lytically permissive cell dies. Herpesviruses solve this problem by limiting the number of infected cells in which the productive phase is activated at any time. In this way, persistence in both the host and the population is ensured. From the host’s standpoint, this strategy also limits pathology. However, what governs the dichotomy between cells that are refractory versus those that are permissive to activating the viral lytic switch, propelling a herpesvirus into its productive phase? Despite great strides in identifying viral and host regulators of the quiescent and productive phases, a central unifying mechanism underlying this seemingly stochastic cell and viral fate decision remained elusive.

There is a rich literature describing viral and host signaling pathways and transcription factors that regulate activation of the lytic phase from latency, i.e., (re)activation of the Epstein–Barr virus (EBV; Human herpesvirus 4). EBV is the prototypic human cancer-causing herpesvirus that is nearly ubiquitous in humans [1,2]. Among its noteworthy properties, is EBV’s ability to transform B lymphocytes and establish latency in culture. EBV in these lymphoblastoid cell lines (LCL) and in explanted Burkitt lymphoma (BL) cell lines remains tightly latent but can be induced by chemical and immune triggers to switch into the lytic phase. Remarkably, EBV in these cells is partially permissive regardless of the lytic trigger. Indeed, the reasons for EBV’s ability to differentially reactivate under different conditions has been a long-standing question with relevance also to several diseases, including transplant- and other immunosuppression-related lymphoproliferative diseases and lymphomas. On the one hand, with the lytic phase playing an essential role in development of lymphoid and epithelial EBV cancers [3,4], such tumors could be prevented by blocking lytic activation. On the other, shifting the balance towards the lytic phase, thereby eliminating the residual subpopulation of the refractory cells, would be a tremendous boost to oncolytic therapies in which latently infected malignant cells are targeted for killing by induction of the viral lytic phase.

## 2. Epigenetic Regulation of the Lytic Phase

In addition to non-coding RNAs, herpesvirus genomes, which persist as multicopy episomes in the nucleus, encode ~90 proteins. While most of these are expressed during the lytic phase, a few, namely nuclear antigens and latent membrane proteins, are expressed during EBV latency. Latency is disrupted upon expression of an immediate-early viral lytic gene, whose product functions as the lytic switch. Gammaherpesviruses encode two lytic switch proteins: ZEBRA/Zta (BamHI Z Epstein–Barr virus replication activator) and RTA (replication and transcription activator). ZEBRA is the primary lytic switch for EBV in B lymphocytes, and RTA (expressed by both EBV and Kaposi’s Sarcoma-Associated Herpesvirus (KSHV), the primary lytic switch for KSHV. Once the lytic phase is initiated, it progresses through expression of temporally regulated early and then late lytic genes, many of which are transcriptional targets of the lytic switch proteins [5]. Thus, in an infected cell, the virus can exist in a latent or lytic state, but generally not both.

While expression of the viral lytic switch clearly marks the beginning of the lytic phase, the host is thought to regulate events that favor or oppose expression of the lytic switch. Therefore, in effect, events upstream of the lytic switch govern the lytic–refractory dichotomy. A number of signaling pathways, including PKC, p38 MAPK, and PI3-K/Akt, and transcription factors, including members of the AP-1 family, C/EBP, CREB, Sp1, MEF2D, and HIF-1α, turn on expression of ZEBRA under different experimental conditions [6,7,8,9,10,11,12]. This information, though valuable, is derived from studying different types of latently infected cells, exposing cells to disparate lytic triggers, and most importantly, examining a mixed population of latent and lytic cells. Consequently, our understanding of events upstream of the lytic switch is not only fragmented but also does not apply to every situation involving lytic activation. For example, PKC activation or damage to cellular DNA, while able to trigger some latently infected EBV^+^ cells into the lytic phase, were later deemed to be neither necessary nor sufficient to activate the lytic switch [13,14].

The task of identifying a potentially unifying mechanism that converges on activation of ZEBRA may be approached by separating lytic from refractory subpopulations without genetically altering the virus or the host. This separation utilizes high titer IgG antibodies in the sera of healthy humans persistently infected with EBV [15]. Using this approach, a key experiment, in which sorted refractory cells were tested for their responsiveness over time to several lytic triggers, revealed a gradual return from total non-responsiveness of the refractory population to baseline/parental levels of partial responsiveness [15]. This suggested that epigenetic mechanisms govern the fated decision to respond to a lytic trigger or remain latent and refractory to such stimuli. Subsequently, genome-, transcriptome-, and proteome-wide experiments, using lytic and refractory sorted cells, have revealed several insights [16,17,18], some of which will be described in the following sections. Notably, (i) exposure of latently infected cells to disparate lytic triggers reshapes the cellular transcriptome into a ‘pro-lytic’ state upstream of the expression of ZEBRA [19]; and (ii) an ancient cellular epigenetic silencing machinery that represses foreign genomes, such as those of endogenous retroviruses, also silences herpesvirus lytic gene expression. Indeed, components of this machinery promote the refractory state of both EBV and KSHV [20,21].

## 3. The Constitutive Heterochromatin Machinery Silences Herpesvirus Lytic Genes

Comparison of the B cell transcriptome in sorted lytic and refractory cells revealed that a number of host genes are differentially expressed in the two subpopulations [17,22]. Most notably, the cellular oncoprotein and transcription factor STAT3, several members of the Krüppel-associated box (KRAB) domain–zinc finger protein (ZFP) epigenetic silencing family, and the histone H3K9me3 methyltransferase Set Domain Bifurcated 1 (SETDB1), are expressed at high levels in latent and refractory cells but downregulated in lytic cells. Furthermore, STAT3 regulates the expression of the KRAB–ZFP proteins and SETDB1 [17]. Importantly, the KRAB–ZFP proteins SZF1 and ZNF557, through epigenetic mechanisms, simultaneously silence multiple lytic genes of all kinetic classes (immediate early, early, and late) on the EBV and KSHV genomes to maintain the latent/refractory state (Figure 1) [20,21].

KRAB–ZFPs, with several hundred members, represent the largest family of transcriptional repressors in tetrapods [23,24,25]. These repressors harbor a KRAB domain upstream of an array of 2 to 40 C2H2 zinc fingers. The KRAB domain recruits the transcriptional corepressor KRAB-associated protein 1 (KAP1/TRIM28/TIF1β) and targets it to DNA via zinc finger domains. KAP1 functions as a scaffold for repressive histone-modifying factors, including SETDB1, the NuRD histone deacetylase complex, the heterochromatin amplification factor HP1 that recognizes H3K9me3, and DNA methyltransferase 3A (DNMT3A) or DNMT3B [26]. Collectively, these factors promote gene silencing (Figure 1). While this KRAB–KAP1 machinery induces constitutive heterochromatin, marked by H3K9me3 histones, such heterochromatic silencing can spread over large genomic distances, likely via HP1-mediated recruitment of SETDB1 that can further propagate H3K9me3 marks.

Downstream of the KRAB–KAP1-mediated heterochromatinization, DNA methylation can lock repressive marks on the endogenous retroviral elements so that expression from such foreign DNA is suppressed despite proliferation and differentiation of cells during embryogenesis [27]. Indeed, CpG methylation can make such silencing heritable, particularly if initiated in early embryogenesis [26,28,29]. While we know that EBV lytic genes are also silenced via CpG methylation during latency [30], the extent to which such methylation follows KRAB–KAP1-mediated heterochromatic changes at the lytic promoters is not known but would be important to address given the agility with which herpesviruses transition from the latent/episomal state to the lytic phase.

The lytic phase of two other herpesviruses, human cytomegalovirus (CMV) and human simplex virus-1 (HSV-1), is also silenced by the constitutive heterochromatin machinery, although the identities of the KRAB–ZFPs that recruit KAP1 are unknown; moreover, a role for KAP1 in silencing HSV-1 lytic genes remains to be fully explored [31,32]. Thus, the lytic phase of at least four human herpesviruses is regulated by the constitutive heterochromatin machinery, and interventions that disrupt this machinery shift the balance towards the lytic phase.

## 4. Lytic Cycle of EBV Can Be Broadly Divided into Two Phases: Initiation and Amplification

KAP1′s ability to remodel chromatin is primarily regulated by post-translational modifications. It harbors an E3 ligase activity for Small Ubiquitin-like Modifier (SUMO) protein and is subject to constitutive SUMOylation within KAP1 oligomers. SUMOylation creates binding sites on KAP1 for two histone modifiers (CHD3 of the NuRD complex, and SETDB1) that mediate histone deacetylation and H3K9 trimethylation, respectively, consequently causing chromatin condensation and transcriptional repression [33,34]. However, phosphorylation of KAP1 at S824 impairs SUMOylation and antagonizes KAP1′s ability to condense chromatin. Indeed, in response to double-strand DNA breaks in heterochromatin, activated ATM (via phosphorylation of S1981) phosphorylates KAP1, resulting in remodeling, relaxation, and repair of damaged DNA [35,36,37]. In this way, KAP1 facilitates repair of double-strand breaks in heterochromatin.

Modification of the S824 residue of KAP1 also represents a shared mechanism for regulating the lytic cycle of at least three members of the herpesvirus family; yet, the responsible kinases are distinct: viral protein kinase in KSHV [38], mTOR in CMV [32], and ATM in EBV (Figure 2) [14]. In this regard, our own experiments revealed that exposure of EBV-infected cells to lytic cycle-inducing agents does not result in rapid phosphorylation of KAP1. Indeed, KAP1 phosphorylation generally occurs downstream of expression of ZEBRA, i.e., after activation of the lytic switch, and results from a series of events starting with ZEBRA-driven transcriptional activation of the viral protein kinase (vPK), vPK-mediated phosphorylation of ATM at a non-canonical phospho-residue (S2996), and ATM-mediated phosphorylation of KAP1 at S824 [14,39]. Phosphorylated KAP1 then causes further de-repression of the ZEBRA-encoding *BZLF1* gene and several early and late lytic genes (also silenced by the KRAB–KAP1 machinery), thereby amplifying the lytic signal. Thus, lytic activation from latency appears to proceed in two broad stages, initiation (from lytic trigger exposure to expression of ZEBRA; first 9–12 h) and amplification (events downstream of ZEBRA; ~12 h onward). This amplification, achieved through the cooperation of the virus-encoded lytic switch, a viral kinase, a cancer-related cellular kinase, and a cancer-related epigenetic corepressor, ensures that the lytic cascade does not abort but progresses robustly to produce large numbers of infectious virions (Figure 2).

A recent study in corneal epithelial cells now suggests a similar amplification cycle for HSV-1 [40]. As a follow-up to an earlier study demonstrating that ATM was required for robust lytic replication of HSV-1 [41], a recent study found that ATM activation was dependent on the HSV-1 immediate-early protein ICP4 and did not require viral genome replication or the presence of DNA lesions [40]. In the setting of the EBV lytic cycle, vPK was found to directly phosphorylate ATM at S2996, but vPK-unrelated phosphorylation of ATM was also observed at S1981, the residue typically phosphorylated in response to DNA damage [39], and yet DNA damage was not observed in these cells. These results suggested that distortion of DNA caused by binding of ZEBRA, a replication protein and transactivator [42], may be detected as DNA ‘lesions’, leading to phosphorylation of ATM at the canonical serine residue. This idea is supported by loss of (phosphorylated S1981) ATM activation following transfection of DNA-binding domain mutants of ZEBRA [43]. That said, low levels of DNA damage that may escape detection by conventional assays or specific types of DNA lesions may activate ATM, as was noted in experiments using DNA crosslinking agents [44]. Finally, while phosphorylation/activation of ATM appears to be downstream of ZEBRA and participates in the amplification cycle, two caveats exist: (i) ATM can be bypassed when ZEBRA is experimentally overexpressed [44], suggesting that forced expression of high levels of ZEBRA may be able to overcome heterochromatin-mediated silencing of lytic genes, presumably due to its ability to drive transcription of methylated lytic promoters [30]; and (ii) direct activation of ATM by agents such as ROS and chloroquine, even in the absence of DNA damage, can activate the lytic switch, albeit not always robustly (Figure 2) [14,44,45,46].

## 5. The Inflammasome as a Common Upstream Mechanism That Disrupts Heterochromatin-Mediated Silencing to Initiate the Lytic Cycle

The KRAB–KAP1-directed heterochromatin machinery silences lytic genes from all kinetic classes of EBV, including *BZLF1*, and enforces the latent state of several herpesviruses, underscoring its importance in the life cycle of herpesviruses. However, while experimental manipulation disrupting the STAT3–KRAB–KAP1 axis enhanced the expression of ZEBRA and the lytic cascade in response to multiple lytic cycle-inducing agents, such manipulation was not always sufficient to turn on expression of ZEBRA, i.e., disrupt latency. Furthermore, phosphorylation of KAP1, a key event in disrupting this axis, generally occurred downstream of ZEBRA expression, suggesting that alternative mechanisms were responsible for the initial expression of ZEBRA.

A remarkable property of EBV is its ability to respond to a variety of lytic activation-inducing stimuli, including histone deacetylase inhibitors (HDACi); the DNA demethylating agent 5-azacytidine; B cell receptor crosslinking; transforming growth factor β; proteasome inhibitors, such as bortezomib; radiation; chemotherapy; ROS; and DNA damage [47,48,49,50,51,52]. A common feature of several of these lytic activation stimuli is their ability to activate the inflammasome [53,54,55,56,57]. Furthermore, with an increasing appreciation of links between inflammasome activation and cancer, and the constant engagement of the inflammasome by threats to the cell, we performed a screen to identify the inflammasome sensors able to regulate the EBV lytic switch. Of the four sensors identified, depletion of NLRP3 was most effective at impairing the expression of ZEBRA [58].

NLRP3 is the most promiscuous inflammasome that can be activated by a myriad cell intrinsic and cell extrinsic danger signals, such as metabolic changes and infectious agents. Upon sensing danger, NLRP3 assembles via the apoptosis-associated speck-like (ASC) adapter protein with pro-caspase 1. This assembly/activation cleaves pro-caspase 1 to activate it, following which active caspase 1 cleaves pro-inflammatory molecules IL-1β, IL-18, and Gasdermin D to activate them [57].

Upon exposure of latently infected cells to all tested lytic triggers, including high glucose as a metabolic trigger, we found thioredoxin-interacting protein (TXNIP), a key inflammasome intermediary, to be upregulated via transcription and released from its binding partner thioredoxin, precipitating NLRP3 assembly and activation. Activation of pro-caspase 1, though unable to induce pro-inflammatory cytokine processing, cleaves and thereby depletes KAP1 in a subpopulation of cells (Figure 2 and Figure 3). Remarkably, depletion in the context of inflammasome activation, instead of phosphorylation of KAP1, is sufficient to activate expression of *BZLF1*, triggering entry of EBV in this TXNIP^hi^KAP1^lo^ subpopulation into the lytic phase (Figure 3). Of note, KAP1 levels decline by no more than 30–50% of baseline, presumably because KAP1 is essential for the survival of the cell. Where and how caspase-1 destabilization of KAP1 occurs has yet to be determined. As NLRP3 is a cytoplasmic inflammasome, caspase-1 activation also occurs in the cytoplasm. Activated caspase-1 may cleave newly made or recycled KAP1, depleting its supply to the nucleus, or travel to the nucleus to cleave free or chromatin-bound KAP1. In this way, EBV has tied its replication to a cellular sensing system that is regularly activated by alterations in metabolic states and danger signals [58]. Thus, when the cell perceives intrinsic or extrinsic threats, EBV is able to reactivate from quiescence, propagate itself, and leave.

## 6. The Constitutive Heterochromatin Machinery Is Able to Differentiate between Self and Foreign Genomes

By exploiting the inflammasome to sense danger and using the activation of the inflammasome to turn on its own replication, EBV reveals a surprising connection between the inflammasome and the epigenome. The KRAB–KAP1 machinery heterochromatinizes pericentromeric repeat regions to keep them silent. Dysregulation at these sites can induce aberrant DNA repair, recombination, missegregation, and even transposon activation, all detrimental to the host. It is perhaps not surprising that the host uses the same machinery to tightly regulate reactivation of persistent foreign genomes. Yet, how does this machinery distinguish between self and foreign targets? Is it even necessary to distinguish between self and foreign DNA? It turns out that SZF1, a KRAB–ZFP that recruits KAP1 to host and gammaherpesvirus DNA, recognizes self, i.e., pericentromeric regions, using a repeat sequence-bearing motif, while targeting EBV lytic genes via non-consensus binding sites. Remarkably, SZF1 does not use the motif to recognize the EBV genome nor the binding sites on the viral genome to silence host genes during latency [16]. This differential approach towards target site recognition reflects a strategy by which the host (a) rapidly silences newly arrived foreign genomes and (b) regulates genomes of persistent invaders without jeopardizing its own genomic integrity.

While the KRAB–KAP1 machinery clearly silences herpesvirus genomes, how these heterochromatin-inducing factors are recruited to the genomes of each of the herpesviruses is not entirely clear. For EBV, we know that SZF1 and ZNF557 both target lytic genes; however, ZNF557 appears to utilize KAP1-independent mechanisms for silencing [21]. Silencing of the KSHV lytic program not only involves the KRAB–ZFPs SZF1 and ZNF557 [21] but also host Nrf2-mediated recruitment of KAP1 and viral LANA to lytic genes [59,60]. How the silencing machinery recognizes CMV and HSV-1 genomes is not presently clear.

## 7. Insights into Genetic Immune Disorders, Metabolic Syndromes, and Herpesvirus Diseases

Studies of the mechanisms that underlie silencing via constitutive heterochromatin also resulted in discoveries related to the biology of EBV in specific groups of patients. For example, with low levels of STAT3 shifting the latent–lytic balance towards the lytic state, and STAT3 transcriptionally regulating SZF1 levels, it was not surprising to find that patients with the primary immunodeficiency disease, known as Job’s syndrome (or autosomal dominant hyper-IgE syndrome), who have a dominant negative mutation in their STAT3 gene, exhibit higher levels of spontaneously lytic cells. This, together with poor T cell control of EBV, likely accounts for higher EBV loads in peripheral blood of Job’s syndrome patients [17,61]. Similarly, the lytic switch is spontaneously turned on in B cells from patients with another genetic disorder known as neonatal onset multisystem inflammatory disease (NOMID). Patients with NOMID exhibit hyperactive inflammatory responses due to an activating mutation in the NLRP3 gene. With the inflammasome constitutively active, NOMID B cells are defective in controlling EBV [58].

Investigation into the relationship between the inflammasome and heterochromatic silencing of EBV also provides mechanistic insights into medical observations related to three human herpesviruses: (1) Diabetes, which frequently activates the NLRP3 inflammasome [62,63], was found to substantially increase the risk of development of EBV-lymphomas in renal transplant recipients in a large Australian study [64]. The recent observation that high glucose is able to activate the EBV lytic switch via the NLRP3 inflammasome [58] and that increased lytic activation predisposes the development of EBV-lymphomas [3] together provide a mechanistic basis for this important clinical observation. (2) Recent studies have reported that diabetes increases the risk and severity of zoster and post-herpetic neuralgia [65,66], both of which are caused by the lytic phase of the varicella–zoster virus (VZV), another human herpesvirus. Again, activation of the inflammasome is the likely link between diabetes and VZV. (3) The long-standing observation that HSV-1 reactivation in the form of cold sores frequently occurs at sites of trauma in the oral mucosa [67] is also likely linked to injury-mediated inflammasome activation at the site.

The high prevalence of endemic Burkitt lymphoma seen in children in Equatorial Africa has long been suspected to be the consequence of a connection between EBV and another infectious agent, the malaria parasite *Plasmodium falciparum*. In this regard, EBV is considered causal to all endemic Burkitt lymphoma in African children. This link between EBV and endemic Burkitt lymphoma, while suspected to be in part a consequence of malaria-induced immunosuppression as well as induction of activation-induced cytidine deaminase in B cells [68], may be further compounded by high EBV loads caused by the routine non-prescription use of the antimalarial drug chloroquine, which activates the EBV lytic cycle by activation of ATM and ATM-mediated phosphorylation and inactivation of KAP1, resulting in dysregulation of heterochromatin [14].

## 8. Concluding Remarks

Constitutive KRAB–KAP1 heterochromatin machinery enforces the latent state, while its deactivation is essential to both disruption of latency as well as amplification of the lytic cascade. This mechanism is functional in culture systems and in the blood of patients with EBV-mediated diseases, and its deactivation by the inflammasome in response to numerous lytic triggers provides an explanation for the stochasticity of the lytic response. Additional mechanisms likely add to the decision of a herpesvirus to respond to lytic triggers. That said, EBV’s use of a cellular defense mechanism as a security system to alert itself of threats to its cellular home, as well as to flip the virus-encoded lytic switch so that the virus may replicate to make its getaway, presents the first example of an infectious agent able to actively exploit the inflammasome to turn on its replication. This further points to the NLRP3 inflammasome as an attractive target to thwart EBV reactivation. In this regard, several NLRP3 inhibitors are undergoing preclinical, phase I, and phase II trials for rheumatologic diseases (https://cen.acs.org/pharmaceuticals/drug-discovery/Could-an-NLRP3-inhibitor-be-the-one-drug-to-conquer-common-diseases/98/i7 Accessed on 11 April 2021).

An important unanswered question is whether eviction of the heterochromatin silencing machinery is sufficient to activate the *BZLF1* promoter. Removing the repressive effect at the *BZLF1* promoter may serve as the trigger if transactivators are already poised at the promoter (i.e., if the lytic phase is the default state such that removal of the silencer turns it on). Alternately, activators of the *BZLF1* promoter may yet be required to be recruited upon loss of the silencing effect. Addressing this question will require a priori knowledge of the identity of cells poised to support the lytic phase—TXNIP^hi^KAP1^lo^ cells represent one such group of pro-lytic cells.

## Figures and Tables

**Figure 1 viruses-13-00846-f001:**
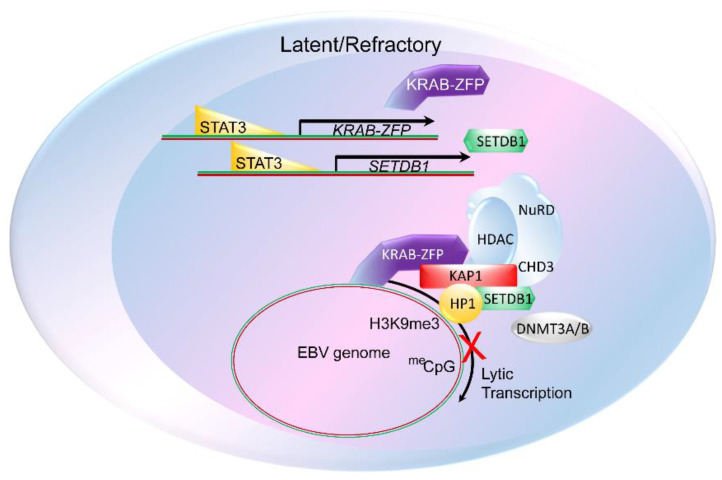
STAT3-mediated regulation of heterochromatin factors and KRAB–KAP1-mediated silencing of the EBV genome in latency. An EBV-infected B cell is illustrated showing the cellular oncoprotein and transcription factor STAT3 transactivating the key heterochromatin regulatory components, including members of the Krüppel-associated box (KRAB) domain–zinc finger protein (ZFP) epigenetic silencing family (KRAB–ZFP), and the histone H3K9me3 methyltransferase, Set Domain Bifurcated 1 (SETDB1). Below, KRAB–ZFPs bind to the EBV genome as well as to KRAB-associated protein 1 (KAP1), which functions as a scaffold for the recruitment of repressive histone-modifying factors SETDB1, heterochromatin amplification factor HP1 that recognizes H3K9me3, the Nucleosome Remodeling and Deacetylase (NuRD) complex, and DNA methyltransferases 3A or 3B (DNMT3A/3B). Histone deacetylase (HDAC) and Chromodomain Helicase DNA Binding Protein 3 (CHD3) are components of the NuRD complex. DNA methylation (meCpG) also results in further suppression of EBV lytic genes during latency and in B cells that are refractory to triggers of the lytic phase. These chromatin modifying events combine to silence lytic gene transcription, keeping EBV latent.

**Figure 2 viruses-13-00846-f002:**
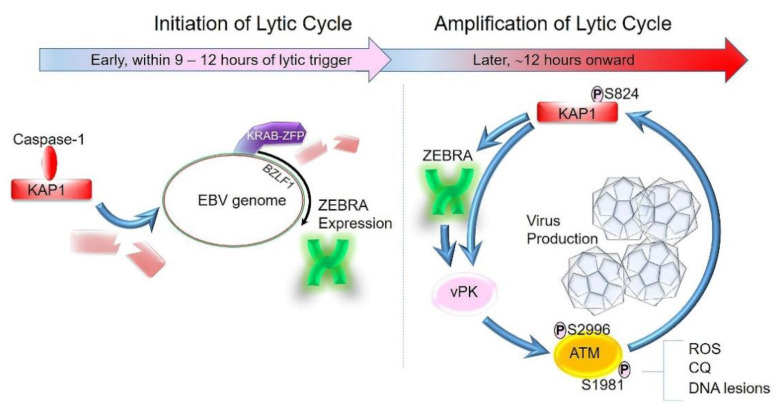
EBV lytic activation from latency proceeds in two stages: initiation and amplification. During initiation, exposure to a lytic trigger/cellular signal results in activation of the inflammasome, thereby activating caspase-1 that destabilizes KAP1, resulting in its depletion. Depletion of KAP1, a key regulator of heterochromatin, allows cellular transcription factors to activate expression of *BZLF1*, which encodes the lytic switch protein ZEBRA, a viral bZIP transcriptional activator (green dimer). These early events within 9 to 12 h of lytic stimulus are followed by an amplification stage in which ZEBRA participates in transactivating its own promoter as well as the promoters of other lytic genes, including replication and transcription activator (RTA). While reactive oxygen species (ROS), DNA damage, or chloroquine (CQ) can activate ATM via phosphorylation at Serine 1981 (S1981), ZEBRA-driven expression of the viral protein kinase (vPK) leads to vPK-mediated phosphorylation/activation of ATM at a non-canonical phospho-residue, S2996. Activated ATM then phosphorylates KAP1 at S824, deactivating KAP1 and antagonizing its ability to condense chromatin. This causes further de-repression of *BZLF1* as well as other early and late lytic genes (also silenced by the KRAB–KAP1 machinery), thereby amplifying the lytic cascade.

**Figure 3 viruses-13-00846-f003:**
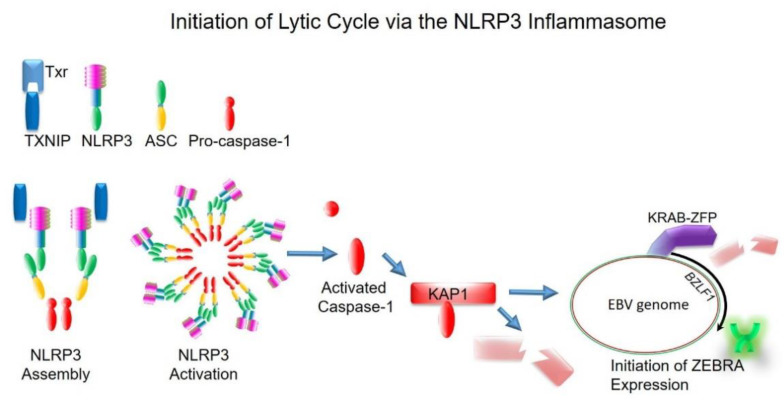
Model depicting NLRP3 inflammasome-mediated activation of the EBV replication switch. Shown are components of the NLRP3 inflammasome, including thioredoxin-interacting protein (TXNIP), the reactive oxygen scavenger thioredoxin (Txr), NLRP3 (NOD-, LRR- and pyrin domain-containing protein 3), apoptosis-associated speck-like (ASC) adapter protein, and pro-caspase-1. Exposure of EBV-infected B cells to a lytic trigger, such as a danger signal, results in upregulation of TXNIP via transcription and/or release from its binding partner Txr, precipitating NLRP3 assembly and activation. NLRP3 assembly/activation involves ASC-mediated recruitment of pro-caspase-1 to NLRP3 oligomers, resulting in intermolecular cleavage of pro-caspase-1 to active caspase-1. Caspase-1 destabilizes the key heterochromatin regulator KAP1, resulting in de-repression of *BZLF1*. ZEBRA (green dimer) transactivates RTA and a multitude of other lytic genes, resulting in productive virus replication.

## Data Availability

Not applicable.

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
