# Peer review of "Inflammasome, the Constitutive Heterochromatin Machinery, and Replication of an Oncogenic Herpesvirus"

_viruses, 2021, doi:10.3390/v13050846_

Round 1

Reviewer 1 Report

The review article "Inflammasome, the constitutive heterochromatin machinery, and replication of an oncogenic herpesvirus" is a very nice, very comprehensive and very clear work addressing key features of the early steps participating to the initiation of the lytic cycle of EBV. It opens and clarifies novel insights describing the inter-relationship existing between the host and the virus. The whole organization of the manuscript is very logical.

I only have one minor concern regarding figure 2. Indeed, in the text (section 4), the left part of the figure is never discussed while figure 2 is called there. The text only mentions the role of KAP-1 phosphorylation but not the role of either caspase-1 or KAP-1 depletion in the initiation of the lytic cycle. Even in the figure 2 legend, caspase-1 is not mentioned. The role of caspase-1 as well as the loss of KAP-1 will only come when Fig 3 is discussed in section 5. I would therefore adapt the text to the figure or the figure to the text to make the story clear.

Author Response

Thank you for your excellent review of our manuscript.

Reviewer 1 noted that the left part of the figure 2 is never discussed in the text (section 4) while figure 2 is called there. Even in the figure 2 legend, caspase-1 is not mentioned. 

Response: we have added descriptive text to the figure legend, introducing KAP1 and caspase-1 roles shown in the left panel of Figure 2.

We now also refer to the left side of Figure 2 in section 5. 

Reviewer 2 Report

Overall this is an excellent review and I have very few concerns with it. The manuscript presents an intriguing idea that EBV can use the inflammasome as a security system to alert itself. This is indeed an excellent example of a virus exploiting the inflammasome to serve its own needs. This notion is becoming a common observation in the virology field.

The model is supported by observations published by this group in PNAS, as well as a Lv/Zhang paper in PLoS Pathog showing that Casp1 can cleave KAP1 to activate EBV replication.

Overall this review provides explanation of a potential link between the inflammasome and the epigenome, which to my knowledge has not yet been established in the context of any other viral infection.

However, perhaps I missed something, but inflammasomes do not assemble in the nucleus, and activated caspase-1 does not go to the nucleus, so where in the cell does the cleavage of KAP1 take place? I think the authors should comment on that gap if it is known.

The Figures are very nicely drawn and there is an extensive coverage of latent/lytic phase regulation of EBV replication.

I applaud the authors for a beautifully written review.

Author Response

Thank you for your excellent review of our manuscript.

Reviewer 2 noted that inflammasomes do not assemble in the nucleus, and activated caspase-1 does not go to the nucleus, so where in the cell does the cleavage of KAP1 take place? I think the authors should comment on that gap if it is known.

 Response: Indeed, this is an important area of investigation, and we have revised the text of Section 5 (lines 269-273) to include a discussion of this remaining knowledge gap. 

"Where and how caspase-1 destabilization of KAP1 occurs has yet to be determined. As NLRP3 is a cytoplasmic inflammasome, caspase-1 activation also occurs in the cytoplasm. Activated caspase-1 may cleave newly made or recycled KAP1 depleting its supply to the nucleus, or travel to the nucleus to cleave free or chromatin-bound KAP1. "